# Association of Life's Simple 7 lifestyle metric with cardiometabolic disease-free life expectancy in older British men
Qiaoye Wang [1] ✉, Amand Floriaan Schmidt [2,3], Lucy T. Lennon[1], Olia Papacosta[1], Peter H. Whincup[4] & Goya Wannamethee[1]

## Abstract

**Background** Cardiometabolic diseases (CMD), including myocardial infarction, stroke, and type 2 diabetes, are leading causes of disability and mortality globally, particularly for people at an older age. The impact of adhering to the Life's Simple 7 (LS7) on the number of years an individual will live without CMD in older adults remains less studied.

**Methods** This study included a cohort of 2662 British men aged 60–79 years free of CMD at baseline from the British Regional Heart Study (BRHS). Each LS7 factor (BMI, blood pressure, blood glucose, total cholesterol, smoking, physical activity, and diet) was categorized as poor, intermediate, or ideal, and a composite LS7 adherence was determined by summing the number of LS7 ideal levels achieved. Flexible parametric Royston–Parmar proportional-hazards model was applied to estimate CMD-free life expectancy.

**Results** Here we show that compared to men with the lowest LS7 adherence [with 18.42 years (95% CI: 16.93, 19.90) of CMD-free life at age 60], men having an ideal LS7 adherence are estimated to gain an additional 4.37 years (95% CI: 2.95, 5.79) of CMD-free life. The CMD-free life gain benefits are consistent across social class groups of manual and non-manual workers. Among LS7 factors, achieving an ideal physical activity provides the largest CMD-free survival benefit: 4.84 years (95% CI: 3.37, 6.32) of additional CMD-free life compared with the physically inactive group.

**Conclusions** Our study quantifies and highlights the benefits of adhering to the LS7 ideal levels for living a longer life without CMD in older adults.

## Plain language summary

Cardiometabolic diseases, including heart attack, stroke, and type 2 diabetes, are leading causes of disability and deaths globally. To benefit cardiometabolic health, the American Heart Association made a number of recommendations, known as the Life's Simple 7 lifestyle metric, including not smoking, having adequate physical activity, following a healthy diet pattern, and managing healthy body weight, healthy blood pressure, cholesterol, and blood sugar levels. Our study showed that adopting a healthy lifestyle following these recommendations could potentially increase the cardiometabolic disease-free life expectancy by more than four years for British men at age 60, with achieving an ideal physical activity level provided the largest survival benefit. Our findings highlight the need for public health efforts and interventions to support older adults in achieving optimal cardiometabolic health, particularly with regards to physical activity.

Cardiometabolic diseases (CMD), defined herein as myocardial infarction (MI), stroke, and type 2 diabetes (T2D)[1], are leading causes of disability and mortality globally. WHO estimated that CMD accounted for ~30% of the world's total deaths in 2019[2]. The prevalence of CMD has been shown to increase with age, and with the population aging worldwide, CMD are becoming major global health burdens[3–5]. The burdens of CMD are directly associated with increased frailty, disability, and mortality in affected individuals, which also translate to significantly higher overall healthcare costs[6,7].

Lifestyle factors are crucial modifiable risk factors for CMD. Specific lifestyle risk factors, such as obesity, physical inactivity, and smoking, have been extensively studied and are associated with the development of CMD[8]. While each risk factor is important individually, it was suggested that lifestyle factors are inter-related and a combined healthy lifestyle may have synergistic effects in preventing diseases[9,10]. There has been more research focusing on understanding the health impacts of a composite lifestyle, which is derived by summing up individual lifestyle metrics such as body mass index (BMI),

[1]Department of Primary Care and Population Health, Institute of Epidemiology and Health Care, University College London, London, UK. [2]Department of Population Science and Experimental Medicine, Institute of Cardiovascular Science, University College London, London, UK. [3]Department of Cardiology, Amsterdam Cardiovascular Sciences, Amsterdam University Medical Centre, University of Amsterdam, Amsterdam, The Netherlands. [4]Population Health Research Institute, St George's University of London, London, UK. ✉e-mail: qiaoye.wang.21@ucl.ac.uk

physical activity level, and smoking status. Among composite lifestyle measures, the Life's Simple 7 (LS7) lifestyle metric developed by the American Heart Association (AHA) in 2010 has been widely studied[11]. The LS7 metric is composed of seven lifestyle-related factors: BMI, blood pressure (BP), blood glucose, total cholesterol, smoking, physical activity, and diet. Prior studies have consistently found the inverse association of LS7 with risk of CMD. A meta-analysis including 11 cohort studies showed that an ideal LS7 profile (i.e., lower BMI, BP, blood glucose, and total cholesterol, regular physical activity, healthy diet, and no smoking) was significantly related to lower risks of incident MI and stroke [HR (95% CI): 0.24 (0.15, 0.34) and 0.33 (0.21, 0.45) for MI and stroke, respectively] when compared to poor LS7 profile[12]. In studies exploring the association of LS7 with risk of T2D, a substantially lower risk of T2D in those with ideal LS7 profile was also observed[13–16].

Most previous research demonstrated the importance of adhering to the LS7 lifestyle metric using relative measures, such as hazard ratios (HRs)[12–16]. While such ratio effect measures might be large, they often lack implications on benefits in absolute risks. Fewer studies have investigated or quantified the impact of LS7 on disease-free life expectancy or years of life gained without chronic diseases, which could capture both the quantity and relative quality of life. Several previous studies demonstrated that a composite healthy lifestyle was related to ~5–12 additional years of life without major chronic diseases[17–20]. However, most of the existing studies focused on middle-aged or younger populations, which did not demonstrate the potential diminished impacts of a healthy lifestyle on disease-free life expectancy in older adults due to age-related physiological changes[20,21]. Meanwhile, very few prior studies have specifically examined years of life gained without CMD, which included three leading chronic diseases in older adults. Moreover, although social disparities have been demonstrated in both adherence to a healthy lifestyle and the risk of CMD[22–24], whether the beneficial impacts of a healthy lifestyle on cardiometabolic health would be modified by social class remain less studied.

In the current study, we set out to quantify the amount of CMD-free life years gained attributable to adherence to LS7 lifestyle metric in older adults, and if social class modifies the association. Specifically, we leverage prospective data from 2662 participants of the British Regional Heart Study (BRHS) applying the Royston–Parmar proportional-hazards model taking both the occurrence of fatal or nonfatal CMD and potential competing risk by deaths of any other causes into account.

Here, we show that closely adhering to the AHA LS7 lifestyle metric is associated with four additional years of CMD-free life for British men at age 60, compared to men with the lowest LS7 adherence. This benefit is shown to be consistent across social class groups of manual and non-manual workers. Additionally, we find that among LS7 factors, having an ideal level of physical activity (i.e., maintaining a usual pattern of moderate to vigorous physical activity level) provides the largest gain in survival free of CMD compared to those being physically inactive.

## Methods
### Study design
The BRHS randomly recruited 7735 men aged 40–59 years from 24 primary care practices across Britain between 1978 and 1980[25]. The current study used baseline data collected from 1998 to 2000, which was marked as the 20-year follow-up (Q20) of the BRHS[26]. At Q20 follow-up, 4252 men (aged 60–79 years) completed follow-up questionnaires on sociodemographic, health, medication, and lifestyles, and underwent a lab-based physical examination[25,26]. LS7 metrics used in this study were generated from the Q20 questionnaires and physical examination data. 3167 Participants without prevalent MI, stroke, or T2D (self-reported or screen-detected defined by fasting plasma glucose ≥ 7 mmol/L) at Q20 were available for the current study. We further excluded participants with missing lifestyle or sociodemographic information (n = 505), leaving 2662 men for the complete-case analyses. All participants provided their written informed consents in accordance with the Declaration of Helsinki for participating in the BRHS. Ethical approval for the BRHS was provided by The National Research Ethics Service Committee London–Central (Reference number: MREC/02/2/91).

### Life's Simple 7 metric
BMI, BP, blood glucose, and total cholesterol were measured objectively at the baseline physical examination, while smoking, physical activity, and diet were self-reported. The level of adherence to each LS7 lifestyle metric was categorized as poor, intermediate, or ideal. The AHA criteria were used to define poor, intermediate, and ideal levels for BMI, BP, blood glucose, and total cholesterol[11]. For diet, physical activity, and smoking, adherence levels were classified using BRHS specific cut-offs[27–33] (details in Supplementary Table 1). To assess adherence to a healthy diet, we employed Elderly Dietary Index (EDI), a dietary score developed specifically for older adults[27]. Diet adherence was categorized into poor, intermediate, and ideal by dividing the EDI scores into tertiles. Physical activity levels were categorized into poor (inactive), intermediate (light to occasional), and ideal (moderate to vigorous) based on participants' usual pattern of physical activity, considering the frequency and type (intensity) of the activity[29–32]. Smoking status was categorized by grouping individuals into current smokers (poor), never smokers (ideal), and ex-smokers (intermediate). For ex-smokers, those who had quit for more than 5 years were also considered as having an ideal smoking status[33]. There is no poor blood glucose level in this study because blood glucose was used to determine screen-detected T2D and participants with T2D were excluded from the analyses. After obtaining each factor's adherence level, the number of LS7 ideal levels achieved at baseline was then summed to generate a total LS7 adherence score. Based on the number of ideal levels achieved, we grouped composite LS7 adherence as poor, intermediate, or ideal for men achieving 0–1, 2–3, or 4+ LS7 ideal levels, respectively[34,35].

### Outcome measures
Participants were followed up for CMD, including MI, stroke, and T2D, and mortality until June 1st, 2018. There is no loss to follow-up in this cohort. For each participant, nonfatal CMD were ascertained from ongoing general practitioner reports and biennial reviews of participants' medical records[26]. Death information was obtained from the National Health Service Central Registers in Southport (for England and Wales) and Edinburgh (for Scotland). The *International Classification of Diseases, Ninth Revision* was used to code causes of death. Fatal MI was coded as 410-414, and fatal stroke was coded as 430-438. CMD-free life expectancy is defined as the time in years from age 60 until the occurrence of incident CMD, death, or reaching 100 years old, whichever came first.

### Covariates
Covariates included in the analyses were social class, alcohol intake, and energy intake, obtained from the baseline self-completed questionnaire[36–39]. Social class was classified into manual (e.g., bricklayers, bus conductors, general laborers), non-manual (e.g., engineers, teachers, clerks), or unspecified based on the longest-held position coded by the Registrar General's occupational classification[36]. Alcohol intake was divided into six groups: none, occasional, light, moderate, heavy, and an unspecified group if drinking amount was unavailable[37]. Total energy intake was computed from the baseline food frequency questionnaire and included as a continuous variable[38,39].

### Statistical methods
Participants were categorized into three groups based on their baseline LS7 adherence. Baseline characteristics were summarized using means with standard deviations for continuous variables and frequency with percentage for categorical variables. Pearson's Chi-squared and Kruskal–Wallis test were used to compare baseline characteristics across the three groups for all categorical variables and continuous variables.

Flexible parametric Royston–Parmar proportion-hazards model was used to estimate the HRs and corresponding 95% confidence intervals (95% CIs) of CMD/death, and CMD-free life expectancies (95% CIs), using age as time scale[40]. We initially performed complete-case analyses in this study, which restricted the analyses to 2662 participants who had complete lifestyle and sociodemographic data. Before estimating the amount of

survival years without CMD, we firstly obtained the HRs (95% CIs) of CMD/death, considering both the diagnosis of CMD and death as events, for composite LS7 adherence and individual LS7 metrics. After obtaining the HRs (95% CIs), residual CMD-free life expectancy was then estimated as the area under the survival curve up to 100 years old, conditioning on surviving free of CMD at ages 60–100 years old (1-year intervals). Specifically, to estimate the CMD-free life expectancy, for each LS7 adherence group, an averaged survival curve was obtained based on individual survival curves, and the area under the survival curve was computed. Lastly, years of CMD-free life gained at 60 years old and 95% CIs were computed by the difference between the areas under two survival curves. For example, years of CMD-free life gained at age 60 for individuals having ideal LS7 metrics was calculated as the difference of CMD-free life expectancy between ideal LS7 adherence group and the reference poor LS7 adherence group. Years of life gained without CMD were calculated for both composite LS7 adherence and individual LS7 metrics. All analyses were adjusted for covariates social class, alcohol intake, and energy intake. To investigate whether social class modifies the association between LS7 adherence and years of CMD-free life gained, we also carried out a secondary analysis stratified by social class. Additionally, several sensitivity analyses were conducted to further examine the robustness of our findings. These included an additional adjustment for the National Index of Multiple Deprivation (IMD) quintile, which is a socioeconomic indicator derived from baseline neighborhood-level socioeconomic factors such as income, employment, education, housing, and living environment[41]. We also assessed the impact of additionally adjusting for baseline prevalent cancer status, which may be a result of a poor lifestyle and affect participants' life expectancy without CMD. Besides, to evaluate the potential impact of missing observations, we carried out multiple imputation. Missing variables were imputed through multiple imputation using chained equations method (with 10 imputed sets). Analyses were performed using STATA 17 statistical software (stpm2 command). All reported p-values and 95% CIs were two-sided, and a p-value < 0.05 was considered statistically significant.

### Reporting summary
Further information on research design is available in the Nature Portfolio Reporting Summary linked to this article.

## Results
### Baseline characteristics of study participants
Of 2662 men being included in the analyses, the mean age at enrollment was 68.2 years. During a median follow-up of 19.3 years (Q1, Q3: 18.8, 19.9), 819 (30.8%) participants developed fatal or nonfatal CMD events, and there were 945 (35.5%) individuals died from other causes. Table 1 shows the baseline characteristics of the study population by their baseline LS7 adherence. Men adhered the least to the LS7 were more likely to be manual workers and heavy alcohol drinkers, and had poorer LS7 metrics, including higher blood glucose and total cholesterol. Comparing to participants with complete data, men with missing observations (n = 505) were older, had less energy intake and poorer BMI measures, and more likely to be manual workers and heavy drinkers, but had similar BP, blood glucose, total cholesterol, smoking, and dietary characteristics (Supplementary Table 2).

### LS7 adherence
LS7 adherence was associated with the risk of CMD/death (Fig. 1). Comparing to participants who had poor LS7 adherence at baseline, participants with intermediate or ideal LS7 adherence had a significantly lower relative risk of developing CMD or death [HR (95% CI): 0.73 (0.64, 0.83), 0.58 (0.50, 0.68), p-value < 0.01]. The results remained unchanged following missing data imputation or additional adjustments for the IMD or baseline prevalent cancer status (Supplementary Tables 3–5).

The benefits of living a healthy lifestyle correspond to an attributable increase of 2.53 (95% CI: 1.44, 3.62) additional years of life without CMD for men with intermediate LS7 adherence, and 4.37 (95% CI: 2.95, 5.79) additional years for those with ideal LS7 adherence, compared to men with poor LS7 adherence at baseline (Fig. 2, Supplementary Table 6). Specifically, participants adhering the closest to the LS7 were estimated to have another 22.79 years (95% CI: 21.27, 24.31) of life without CMD at age 60, while those with intermediate or the least LS7 adherence were expected to have 20.95 years (95% CI: 19.64, 22.25) or 18.42 years (95% CI: 16.93, 19.90) of additional CMD-free life (Supplementary Table 6). In the subgroup analysis of association between LS7 adherence and CMD-free life years gained, social class did not modify the association significantly (p-value for interaction = 0.8) (Fig. 3, Supplementary Table 7). Manual workers were estimated to gain 4.46 years (95% CI: 2.38, 6.53) of additional life without CMD if achieving an ideal LS7 adherence, compared to poor LS7 adherence. Non-

**Table 1 | Baseline characteristics of 2662 older British men by baseline adherence to the LS7 lifestyle metric**

| Characteristic | Total | Baseline adherence to the LS7 lifestyle metric | | | |
| --- | --- | --- | --- | --- | --- |
| | | Poor (0–1 ideal levels) | Intermediate (2–3 ideal levels) | Ideal (4+ ideal levels) | P-value |
| No. of participants, n (%) | 2662 (100) | 403 (15.1) | 1630 (61.2) | 629 (23.6) | |
| Incident CMD/death, n (%) | 1764 (66.3) | 309 (76.7) | 1095 (67.2) | 360 (57.2) | <0.01 |
| Male, n (%) | 2662 (100) | 403 (100) | 1630 (100) | 629 (100) | |
| Age at baseline, years | 68.2 (5.4) | 67.9 (5.3) | 68.4 (5.5) | 67.8 (5.3) | 0.1 |
| Manual social class, n (%) | 1216 (45.7) | 238 (59.1) | 749 (46.0) | 229 (36.4) | <0.01 |
| Heavy alcohol intake, n (%) | 75 (2.8) | 18 (4.5) | 46 (2.8) | 11 (1.8) | <0.01 |
| Energy intake, kcal/day | 2152.8 (519.0) | 2192.2 (549.9) | 2144.6 (526.7) | 2148.8 (476.6) | 0.4 |
| BMI, kg/m$^2$ | 26.6 (3.4) | 28.5 (3.4) | 26.8 (3.3) | 24.8 (2.9) | <0.01 |
| Systolic BP, mmHg | 149.6 (23.3) | 153.9 (21.6) | 150.9 (22.7) | 143.4 (24.9) | <0.01 |
| Diastolic BP, mmHg | 86.0 (10.7) | 87.8 (11.0) | 86.4 (10.3) | 83.9 (11.3) | <0.01 |
| Blood glucose, mmol/L | 5.6 (0.5) | 5.9 (0.5) | 5.6 (0.5) | 5.3 (0.4) | <0.01 |
| Total cholesterol, mmol/L | 6.1 (1.1) | 6.4 (0.9) | 6.1 (1.0) | 5.7 (1.1) | <0.01 |
| Physically inactive, n (%) | 232 (8.7) | 74 (18.4) | 139 (8.5) | 19 (3.0) | <0.01 |
| Current smokers, n (%) | 346 (13.0) | 146 (36.2) | 183 (11.2) | 17 (2.7) | <0.01 |
| EDI score | 24.2 (3.2) | 22.5 (2.9) | 23.9 (3.1) | 26.0 (3.0) | <0.01 |

Values are presented as Mean (SD) or n (%) unless stated otherwise. Pearson's chi-squared test was used for all categorical variables. Kruskal–Wallis test was used for all continuous variables.
CMD cardiometabolic disease, LS7 Life's Simple 7, BMI body mass index, BP blood pressure, EDI elderly dietary index.

manual workers would gain 4.12 years (95% CI: 1.88, 6.35) of additional CMD-free life, comparing ideal to poor LS7 adherence.

### Individual LS7 metrics

The relationships between individual LS7 metrics and years of life gained without CMD are presented in Fig. 4 (Supplementary Table 8). The largest CMD-free survival benefit was observed for physical activity: men who achieved an ideal level of physical activity had an estimated 4.84 years (95% CI: 3.37, 6.32) of longer CMD-free life compared to those who had a poor

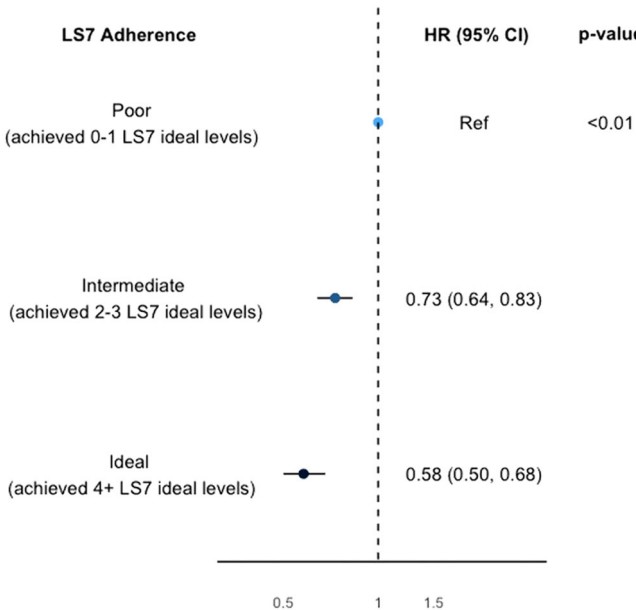

**Fig. 1 |** The risk of cardiometabolic diseases/death according to baseline Life's Simple 7 adherence. Hazard Ratios and 95% Confidence Intervals of cardiometabolic diseases or death during follow-up in BRHS participants aged 60–79 in 1998–2000 ($n$ = 2662), according to their baseline Life's Simple 7 adherence. Model used age as time scale, and adjusted for social class, alcohol intake, and energy intake. CI confidence interval, HR hazard ratio, LS7 Life's Simple 7.

physical activity level. Having an ideal smoking status provided the second largest CMD-disease free benefit, which had an estimated 3.92 years (95% CI: 2.74, 5.10) of longer life free of CMD compared to current smokers at age 60. Although some lifestyle metrics correlated with each other (Supplementary Fig. 1), the correlations were weak and their individual associations with risk of CMD/death did not change after controlling for each other (Supplementary Fig. 2).

### Discussion

The current study aimed to quantify the number of years of life gained without CMD attributable to adherence to the LS7 lifestyle metric in older adults and investigated whether social class modified the association. Our results revealed that, for older British men living an ideal composite lifestyle as recommended by the AHA LS7 lifestyle metrics (i.e., maintaining healthy BMI, BP, blood glucose, and total cholesterol levels, and regularly participating in physical activity, eating a healthy diet, and not smoking), at the age of 60, they could potentially gain more than 4 years of life without CMD, compared to those reporting a poor LS7 adherence. And this health benefit was consistent across social class groups. Notably, among LS7 lifestyle metrics, we found that having a moderate to vigorous physical activity level improved the CMD-free life expectancy the most, with an estimated of 4.84 years of life gain compared to participants who were physically inactive.

Findings of the current study are consistent with previous studies, which suggest that a composite healthier lifestyle is associated with a longer life expectancy without chronic diseases[17–20]. Specifically, a prior study that involved 12 European cohorts computed a composite lifestyle score based on BMI, smoking, physical activity, and alcohol consumption[17]. The study found that individuals with the best composite lifestyle score had an average of 9.9 additional years of life without chronic diseases for men and 9.4 additional years for women between ages 40 and 75 years, compared to those with the worst lifestyle score. Another prior study including two large cohorts in the UK and USA showed that people could gain up to 12 years of life without chronic diseases at the age of 50 if having no behavioral risk factors (i.e., having a less than 30 kg/m² BMI, not smoking, being physically active, and consuming alcohol less than 5 days a week)[20]. Variations in the estimated years of life gained across prior studies and between ours could be mainly due to the differences in the definitions of ideal healthy lifestyles, cohort population characteristics, and chronic conditions included in the

**Fig. 2 | Cardiometabolic disease-free life gained by Life's Simple 7 adherence.** Years of life gained without cardiometabolic diseases for BRHS participants at age 60 by their baseline Life's Simple 7 adherence ($n$ = 2662). Reference group is the poor LS7 adherence (achieved 0–1 LS7 ideal level) group [18.42 years (95% CI: 16.93, 19.90) of CMD-free life at age 60]. Error bars shown in the figure represent the 95% CIs. Models used age as time scale, and adjusted for social class, alcohol intake, and energy intake. Royston–Parmar model survival curves were used to estimate the CMD-free life years gained and 95% CIs. CI confidence interval, CMD cardiometabolic diseases, LS7 Life's Simple 7.

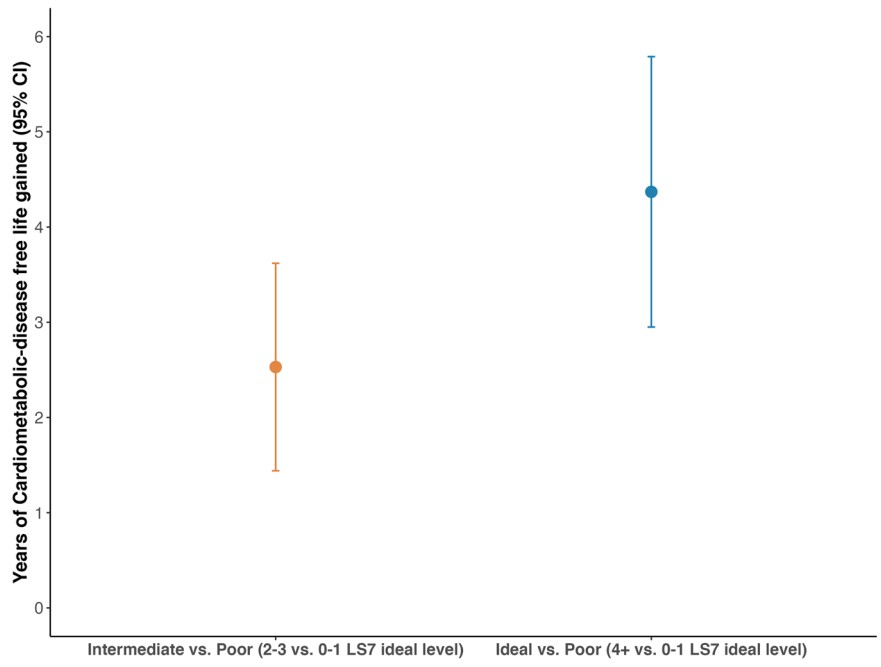

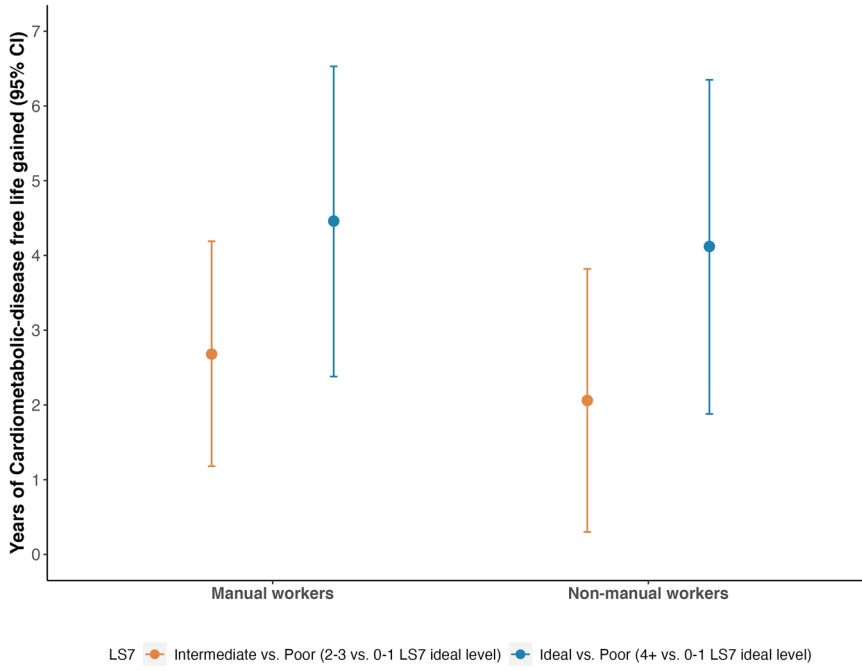

**Fig. 3 | Cardiometabolic disease-free life gained by Life's Simple 7 adherence, stratified by social class.** Years of cardiometabolic disease-free life gained at age 60 by baseline Life's Simple 7 adherence for BRHS manual workers (*n* = 1216) and non-manual workers (*n* = 1346). Reference group is the poor LS7 adherence (achieved 0–1 LS7 ideal level) group. Error bars shown in the figure represent the 95% CIs. Models used age as time scale and adjusted for alcohol intake and energy intake. Royston–Parmar model survival curves were used to estimate the CMD-free life years gained and 95% CIs. CI confidence interval, CMD cardiometabolic diseases, LS7 Life's Simple 7.

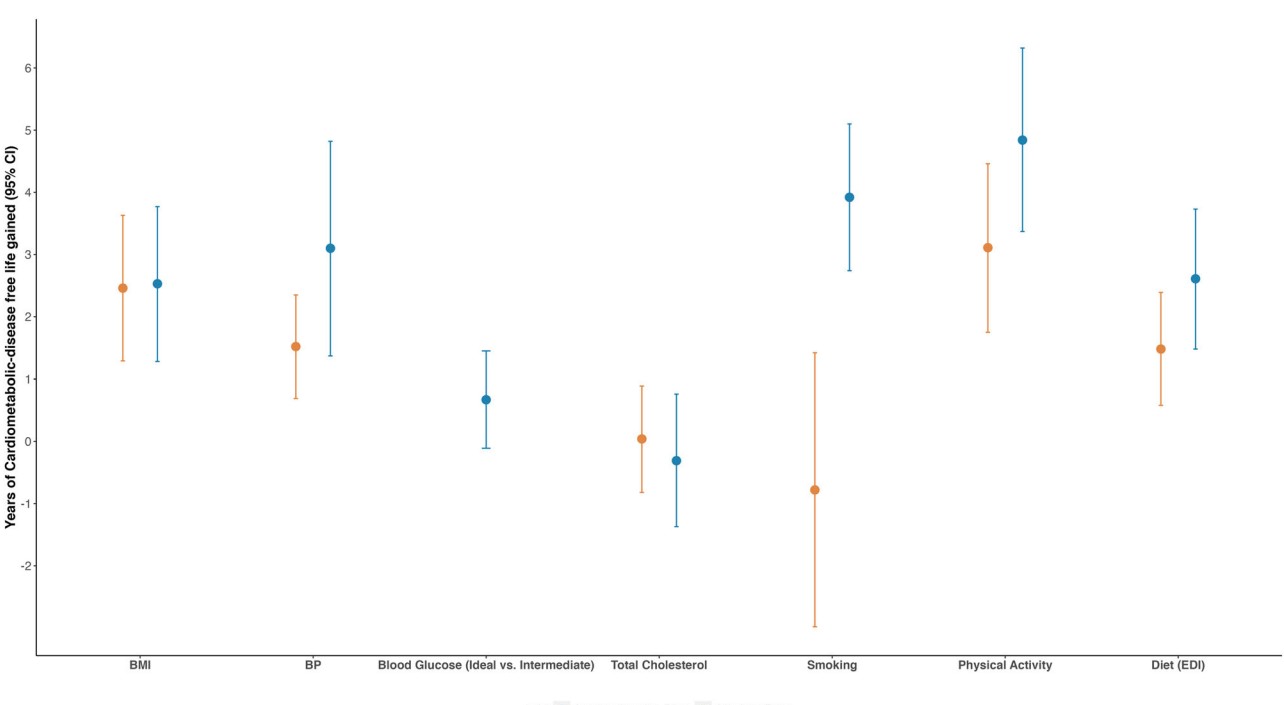

**Fig. 4 | Cardiometabolic disease-free life gained by each Life's Simple 7 metric.** Years of cardiometabolic disease-free life gained at age 60 for BRHS participants by each Life's Simple 7 lifestyle metric (*n* = 2662). Participants with poor glucose status were diabetic and excluded from the analysis. Reference group is the poor group for each LS7 metric. Error bars shown in the figure represent the 95% CIs. Models used age as time scale, and adjusted for social class, alcohol intake, and energy intake. Royston–Parmar model survival curves were used to estimate the CMD-free life years gained and 95% CIs. CI confidence interval, BMI body mass index, BP blood pressure, EDI elderly dietary Index; LS7 Life's Simple 7.

analyses. Results of the current study were most comparable to the findings of a recent UK cohort study using LS7 measures[42]. Following 341,331 participants for a median of 11.4 years, the previous study estimated that UK males without CMD at baseline would gain 4.55 additional years of life at age 45 comparing those with an ideal LS7 profile to those with a poor LS7 profile. Our estimates were a bit smaller mainly because our study involved an older cohort and considered the survival benefits free of CMD. Therefore, while

results of the current study kept with conclusions of prior studies, we extended previous work by quantifying the amount of survival years gained without CMD in older adults, through assessing lifestyles comprehensively with LS7 metric and including the three leading cardiometabolic conditions in combination in the analyses. Our study showed that the CMD-free survival benefit of living an overall healthier life defined by LS7 was quite large for British men at the age of 60, hence confirmed the importance of

pursuing a composite healthy lifestyle to live healthier and longer for UK males even at an older age.

It has been well-documented that manual workers have higher risk of CMD, higher mortality risk, and lower disease-free life expectancy[23,24,43]. Our study also observed a notable discrepancy in CMD-free life expectancy between manual and non-manual workers. Irrespective of LS7 adherence level, manual workers had ~3 fewer years of CMD-free life compared to non-manual workers (Supplementary Table 7). Potential explanations of the health disparity between manual and non-manual workers are that manual workers may have less favorable working conditions, limited healthcare access, and higher stress levels[44]. This social class difference in healthy life expectancy warrants further clinical and public health consideration. However, although manual workers were estimated to have fewer years of CMD-free life expectancy, they could potentially gain similar or even slightly more years of CMD-free life if adopting a healthy lifestyle. These results, while highlighted the existing health disparities across social class groups, further demonstrated the importance of targeted healthy lifestyle promotions and interventions for manual workers. Such interventions have the potential to help this socially vulnerable group of people live a longer and healthier life.

Our study also examined the effects of individual LS7 metrics on CMD-free life expectancy, and we observed different impacts. We found that maintaining a usual pattern of moderate to vigorous physical activity, which indicated an ideal physical activity level, could potentially provide the largest CMD-free survival benefit of all components of the LS7 score. An ideal smoking status (i.e., never smoking or having given up smoking for more than 5 years) provided the second largest number of years life gained without CMD, followed by ideal BP, diet, and BMI levels. Achieving ideal blood glucose and total cholesterol levels did not yield CMD-free survival benefits in the current study. Several prior studies also suggested that among lifestyle metrics, physical activity, and smoking provided larger life expectancy benefits with or without chronic conditions[17,20,41,45,46]. Additionally, though having an ideal smoking status provided a large CMD-free life expectancy benefit, our study suggested that having an intermediate smoking status at baseline, which was defined as giving up smoking for no more than 5 years in our study, did not show CMD-free survival benefit compared to current smoking. But for other individual LS7 metrics that exerted survival benefits, including BMI, BP, physical activity, and diet, gradient benefits were found for intermediate and ideal levels compared to poor levels. Our results, along with previous evidence, further emphasized the importance of engaging in more physical activities and quitting smoking as early as possible.

Our study used a modified LS7 measure, with specific modifications made to the subjective LS7 factors, including diet, physical activity, and smoking status. These modifications aimed to more precisely capture the characteristics and lifestyle impacts on cardiovascular health in the BRHS. Previous BRHS reports have demonstrated associations between the modified LS7 factors and markers or risk of cardiovascular diseases[28,29,33]. The diet measure EDI has been shown to be associated with the risk of CHD in the BRHS[28], and physical activity and smoking status have been found to be related to the markers of cardiovascular diseases in the BRHS[29,33].

Since the introduction of LS7, the AHA has updated and introduced a new Life's Essential 8 (LE8) construct, which added in a new component, sleep health, to the metric[47]. The current study has focused on the LS7 metric as we did not record information on sleep health. Though the LE8 score captures more inter-individual variations, LS7 and LE8 scores were highly correlated in a cross-sectional study of US adult men and women ($\rho = 0.88$, $P < 0.0001$)[48]. A recent prospective study also suggested that there was no distinguishable difference in between the two metrics in their associations with the risk of CVD and that LS7 maybe a more practical lifestyle index due to its ease of data collection and calculation[49]. Our current study, focusing on the simpler metric LS7, showed an association with increased life expectancy free of CMD in older adults. Further studies are warranted to investigate if the LE8 and the addition of sleep health would benefit CMD-free life expectancy even further.

The current study, to the best of our knowledge, is the first to investigate and quantify the effects of LS7 lifestyle metric on life expectancy free of CMD specifically in older adults. Key strengths of the study include a long follow-up term, detailed lifestyle measurements, and precise outcome ascertainments from record reviews. In addition, flexible parametric model used in the analyses enabled us to adjust for several covariates when estimating CMD-free life expectancy, which reduced the potential for confounding bias.

Several limitations of our study need to be considered. First, as in many large prospective studies, several lifestyle measures, including physical activity, smoking, and diet, were quantified through self-reported questionnaires, and this may lead to reporting bias. Second, LS7 measurements were obtained only once at baseline in the study, and potential changes over time were not accounted for. Lifestyle trajectory may be important in affecting disease-free life expectancy as suggested by prior studies[50]. Third, our study only included British men who were mostly from European ethnic origin. The impacts of lifestyle on the CMD-free life expectancy for women and other ethnic groups may be different. Fourth, since our study is an observational study, we did not evaluate if lifestyle intervention, particularly on smoking or physical activity, would provide a longer healthier life free of CMD for older adults as suggested in the current study. Lastly, although analyses were adjusted for known possible sources of bias, due to the study's observational nature, there still could be unmeasured or residual confounding factors.

In conclusion, the current study quantified the likely benefits of adhering to the AHA LS7 lifestyle by estimating the years of life gained free of CMD in British men at the age of 60. We found that achieving 4 or more LS7 ideal levels was associated with a potential gain of 4.37 years of CMD-free life for older British men, and these benefits appeared to be consistent across social class groups. Our study also suggested that among LS7 lifestyle factors, maintaining a usual pattern of moderate to vigorous physical activity level and never smoking or having stopped smoking for more than 5 years may convey the largest gains in CMD-free life expectancy.

## Data availability

The raw data that support the findings of this study are not publicly available due to privacy and confidentiality considerations and are available from the BRHS study manager Lucy T. Lennon (l.lennon@ucl.ac.uk) upon reasonable request. Data are located in controlled access data storage at UCL. Source data for the figures can be found in Supplementary Tables 6–8.

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

## Acknowledgements

The British Regional Heart Study is funded by a British Heart Foundation (BHF) grant (grant number RG/19/4/34452). AFS is supported by BHF grant PG/22/10989, the UCL BHF Research Accelerator AA/18/6/34223, MR/V033867/1, and the National Institute for Health and Care Research University College London Hospitals Biomedical Research Centre. The views expressed in this material are those of the authors and do not necessarily reflect the views of the funding body. This work uses data provided by patients and collected by the National Health Service as part of their care and support. We would like to thank the GP partners, general practices, and participants of the British Regional Heart Study.

## Author contributions

Qiaoye Wang and Goya Wannamethee contributed to the study conception and design; Peter H. Whincup and Lucy T. Lennon planned the data collection; Olia Papacosta contributed to data preparation and data analysis; Qiaoye Wang performed the statistical analysis and prepared the first draft of the manuscript; Amand Floriaan Schmidt and Goya Wannamethee commented and edited the previous versions of the manuscript. All authors critically reviewed and approved the final manuscript.

## Competing interests

The authors declare no competing interests. Amand Floriaan Schmidt is an Editorial Board Member for *Communications Medicine*, but was not involved in the editorial review or peer review, nor in the decision to publish this article.
