## [Peer Review File · Communications Medicine]

Reviewers' comments:

Reviewer #1 (Remarks to the Author):

The authors have done commendable work on this manuscript titled "Association of Life's Simple 7 Lifestyle Metric with Cardiometabolic disease-free life expectancy in Older British men". The methods, results and conclusions are nicely summarized. My biggest worry was limitations, which authors have appropriately written. My only suggestion is to temper down the language for the final conclusion.

Reviewer #2 (Remarks to the Author):

This is a population-based study based on data from the British Regional Heart Study examining the association between Life's Simple 7 and cardiometabolic disease (CMD) free life expectancy. However, the novelty of the study was limited by the fact that a similar article from a large-scale UK cohort has been published (DOI: 10.1001/jamainternmed.2023.0015). Even if the authors had indicated that the population of this study was elderly men, this wouldn't add much to the innovation. Previously published studies have even used larger sample sizes and a more comprehensive assessment of cardiovascular health Life Essential 8, not just Life's Simple 7.

Life's Simple 7 used in the study was modified from the British Regional Heart Study. Does the modification impact the effectiveness of the rating on cardiovascular health? A more in-depth discussion is needed. Also, Life Essential 8 has been recommended to have better assessment validity for cardiovascular health, therefore, why didn't the authors consider the effects of sleep?

The analysis only focused on CMD-free participants. It is well known that a variety of chronic diseases occur simultaneously and mutually reinforcing in the elderly and can contribute to impairing life expectancy. However, this study did not take into account other chronic diseases, such as dementia, cancer, chronic kidney disease, and so on.

Only social class, alcohol intake, and energy intake were adjusted in the model, but some important confounders such as education level, income, and other chronic diseases, which are associated with life expectancy should be controlled for.

Reviewer #3 (Remarks to the Author):

The manuscript titled "Association of Life's Simple 7 lifestyle metric with cardiometabolic disease-free life expectancy in older British men" investigates whether a composite lifestyle measure in older men is associated with risk and years of life free from cardiometabolic disease and death. This composite measure is one of many, which captures the participants level of adherence to 7 select lifestyle factors, ranging from smoking status, to physical activity and BMI. Here, the authors found that older men adhering to an 'ideal' lifestyle gained more years of CMD-free life than men who adhered to a 'poor' lifestyle. The authors also focused on each factor of the composite measure individually, and found physical activity to provide the greatest benefit in CMD-free survival compared to those men that were more inactive.

This was a well written study that confirms our general understanding of these lifestyle factors. For example, being physically active is better for your health than being inactive. While this is of value to the knowledge base, the authors description and interpretation of the composite measure weakens the importance of these findings. Interpreting such composite measures is a challenge, but is vitally important to what the reader can extract from the current evidence. My recommendation is to have the authors revise this aspect of the paper before it can be considered for publication. My specific comments are as following:

ABSTRACT

1. What is the Life's Simple 7? Can the authors provide a brief comment regarding the factors included in the LS7, as this will help make sense of measures categories i.e., poor, intermediate or ideal, and the significance of these results
2. Line 38: Can the authors include the percentage of participants with fatal or non-fatal CMD events e.g., 819 (31%)
3. Line 41: "life gain of 4.37..... compared to men with the least Ls7 adherence" Can the authors include the number of years for those with the least LS7 adherence.

INTRODUCTION

1. Line 79: "Lifestyle factors are crucial modifiable risk factors for CMD..." can the authors provide a brief example e.g., physical activity reduces risk, and how this can be used to derive measures of overall lifestyle e.g., these can be used to derive measures of overall lifestyle.
2. Line 78: "Among lifestyle measures, the Cardiovascular Health (CVH)..... has been widely studied" I recommend the authors stick with CVH or LS7, not both.
3. Line 85: "that an ideal CVH profile" can the authors give an example of what this entails e.g., "an idea CVH profile (i.e., lower BMI, blood pressure and greater physical activity) was significantly related to a lower risk of incident MI and stroke..."

METHODS:

1. Line 132: "To reflect adherence to the LS7 metric criteria, each LS7 metric was categorized as poor, intermediate, or ideal". Are the authors able to link this sentence to the 7 factors categorized by the level of adherence. For example, "The level of adherence to each of these 7 lifestyle metrics was defined as poor, intermediate or ideal..."
2. Lines 55-56: can the authors provide an example for each social class? For example, "non-manual (e.g., office work)...."

RESULTS:

1. Line 197-198: are the authors able to provide a breakdown of the CMD and death events, as per the abstract i.e., "819 fatal or non-fatal CMD events occurred, as well as 39 945 individuals who died from other causes."
2. Line 199: given a LS7 category of poor adherence is based on measures of high blood pressure, low physical activity, smoking and a higher BMI, we would expect these findings, and thus I see no real purpose in reporting these metrics. However, what the authors have written regarding work, alcohol, blood glucose and total cholesterol are interesting and should remain.

DISCUSSION:

1. At the start of the discussion can the authors begin by reinstating what their primary aims of the study were, followed by an interpretation of their findings. Currently it reads like the results.
2. Line 252: "Findings of the current study are consistent with previous studies" can the authors include references here.

3. Line 262: "...UK cohort study. Followed 341,331..." can the authors revise the start of this sentence e.g., "Following 341,331 participants for a median of 11.4 years...."
4. Line 252-272: can the authors elaborate on the type of factors included in these composite lifestyle measures. While the last reference is most similar to this study, it would be interesting to get some general understanding of what type of composites equated some of these other results.
5. Line 274-284: can the authors provide a brief explanation as to why they think manual workers may have fewer years of CMD-free life compared to non-manual workers? What does the evidence suggest?
6. Line 287: "ideal physical activity level" can the authors note what this means e.g., moderate activity per week etc...

Response to Referees

Reviewer #1 (Remarks to the Author):

The authors have done commendable work on this manuscript titled "Association of Life's Simple 7 Lifestyle Metric with Cardiometabolic disease-free life expectancy in Older British men". The methods, results and conclusions are nicely summarized. My biggest worry was limitations, which authors have appropriately written. My only suggestion is to temper down the language for the final conclusion.

Response: We thank the reviewer for the suggestion of tempering down the language for the final conclusion. We have revised the final conclusion paragraph to a more measured and cautious tone:

“In conclusion, the current study quantified the likely benefits of adhering to the AHA LS7 lifestyle by estimating the years of life gained free of CMD in British men at the age of 60. We found that achieving 4 or more LS7 ideal levels was associated with a potential gain of 4.37 years of CMD-free life for older British men, and these benefits appeared to be consistent across social class groups. Our study also suggested that among LS7 lifestyle factors, maintaining a usual pattern of moderate to vigorous physical activity level and never smoking or having stopped smoking for more than 5 years may convey the largest gains in CMD-free life expectancy.”

Reviewer #2 (Remarks to the Author):

This is a population-based study based on data from the British Regional Heart Study examining the association between Life's Simple 7 and cardiometabolic disease (CMD) free life expectancy. However, the novelty of the study was limited by the fact that a similar article from a large-scale UK cohort has been published (DOI: 10.1001/jamainternmed.2023.0015). Even if the authors had indicated that the population of this study was elderly men, this wouldn't add much to the innovation. Previously published studies have even used larger sample sizes and a more comprehensive assessment of cardiovascular health Life Essential 8, not just Life's Simple 7.

Response: We thank the reviewer for raising the issues of innovation and the use of LS7 in our study.

While acknowledging the existence of current comparable studies and appreciating the reference the reviewer provided, we would like to emphasize the unique contributions of our study. Our research focused on a representative cohort of older British men, all aged 60-79 years, recruited from 24 primary care practices across Britain. In contrast, the referenced study used the UK Biobank cohort, which only includes adults up to 69 years old. Our cohort, hence different from other studies, includes a substantial proportion of the elderly, which are a group of individuals who are often considered as being at high risk for cardiometabolic diseases. Previous evidence has also suggested that the beneficial impact of a healthy lifestyle tends to decrease with increasing age²¹. Thus, our study specifically addresses the question of to what extent the benefits of a healthy lifestyle would persist within the elderly, providing valuable additional insights into the impact of lifestyle in this specific high-risk demographic group. Besides, our study highlights the health benefits of a healthy lifestyle specifically

through the lens of cardiometabolic health, providing a distinctive perspective compared to general chronic disease patterns.

We recognize the recently proposed LE8, which added a sleep health score to LS7 and expanded the levels of LS7 health measure domains. Unfortunately, the BRHS study did not record information on sleep at age 60-79 years, so that we were unable to derive the complete LE8 score. Nevertheless, an AHA survey has shown that the LS7 and LE8 were highly correlated (correlation coefficient $\rho = 0.88$, $P < 0.0001$)⁴⁸. Another recent study also suggested that there was practically no distinguishable difference between LS7 and LE8 in their associations with the risk CVD, and LS7 may be a more practical lifestyle index due to its ease of data collection and calculation⁴⁹. Our current study focused on the simpler metric LS7. Given the comparability between the two measures^{48,49}, we would expect that findings based on LS7 would tend to be replicable for LE8. Nevertheless, it warrants further investigation. We have now added a discussion paragraph on LE8, lines 340-350:

“Since the introduction of LS7, the AHA has updated and introduced a new Life’s Essential 8 (LE8) construct, which added in a new component, sleep health, to the metric⁴⁷. The current study has focused on the LS7 metric as we did not record information on sleep health. Though the LE8 score captures more inter-individual variations, LS7 and LE8 scores were highly correlated in a cross-sectional study of US adult men and women ($\rho = 0.88$, $P < 0.0001$)⁴⁸. A recent prospective study also suggested that there was no distinguishable difference in between the two metrics in their associations with the risk of CVD, and that LS7 maybe a more practical lifestyle index due to its ease of data collection and calculation⁴⁹. Our current study, focusing on the simpler metric LS7, showed a significant impact on life expectancy free of CMD for older adults. Further studies are warranted to investigate if the LE8 and the addition of sleep health would benefit CMD-free life expectancy even further.”

Manuscript References:

21. Tsai, M.-C., Lee, C.-C., Liu, S.-C., Tseng, P.-J. & Chien, K.-L. Combined healthy lifestyle factors are more beneficial in reducing cardiovascular disease in younger adults: a meta-analysis of prospective cohort studies. *Sci Rep.* **10**, 18165 (2020).
48. Lloyd-Jones D. M. *et al.* Status of Cardiovascular Health in US Adults and Children Using the American Heart Association’s New “Life’s Essential 8” Metrics: Prevalence Estimates From the National Health and Nutrition Examination Survey (NHANES), 2013 Through 2018. *Circulation.* **146**, 822–835 (2022).
49. Howard, G. *et al.* Comparative Discrimination of Life’s Simple 7 and Life’s Essential 8 to Stratify Cardiovascular Risk: Is the Added Complexity Worth It? *Circulation.* **148**, 00-00. CIRCULATIONAHA.123.065472 (2023).

Life’s Simple 7 used in the study was modified from the British Regional Heart Study. Does the modification impact the effectiveness of the rating on cardiovascular health? A more in-depth discussion is needed. Also, Life Essential 8 has been recommended to have better assessment validity for cardiovascular health, therefore, why didn't the authors consider the effects of sleep?

Response: Our study modified the AHA defined LS7 adherence in the three subjective measures: diet, physical activity, and smoking status. Specifically, we employed Elderly Dietary Index (EDI), a dietary score that was developed specifically for older adults, to measure adherence to a healthy diet²⁷. Diet adherence was categorized as poor, intermediate, and ideal by dividing the EDI score into tertiles. For physical activity, based on a score

derived from participants' usual pattern of physical activity that considered frequency and type (intensity) of the activity, we categorized physical activity into poor (inactive), intermediate (light to occasional), and ideal (moderate to vigorous)²⁹. For smoking status, current smokers were classified as having poor smoking status, never smokers were classified as ideal, ex-smokers were grouped as intermediate but re-classified as ideal if had quit for more than 5 years³³.

These modified LS7 measures used in the current study were designed to be more specific and precisely capture the cardiovascular health rating of the older UK population. Previous BRHS reports have demonstrated and validated the effectiveness of using these measures to assess cardiovascular health in the BRHS^{28,29,33}. Specifically, the diet score EDI has been shown to be associated with risk of CHD in the BRHS²⁸. Physical activity and smoking status were found to be significantly related to the markers of cardiovascular diseases in the BRHS^{29,33}.

To further elaborate on the modified LS7 measures used in the current study and their effectiveness in rating cardiovascular health in the BRHS, we have included some additional explanations in the methods section, lines 140-149, and a discussion paragraph in lines 332-338:

Lines 140-149:

“For diet, physical activity, and smoking, adherence levels were classified using BRHS specific cut-offs²⁷⁻³³ (details in Supplementary Table 1). To assess adherence to a healthy diet, we employed Elderly Dietary Index (EDI), a dietary score developed specifically for older adults²⁷. Diet adherence was categorized into poor, intermediate, and ideal by dividing the EDI scores into tertiles. Physical activity levels were categorized into poor (inactive), intermediate (light to occasional), and ideal (moderate to vigorous) based on participants' usual pattern of physical activity, considering the frequency and type (intensity) of the activity²⁹⁻³². Smoking status was categorized by grouping individuals into current smokers (poor), never smokers (ideal), and ex-smokers (intermediate). For ex-smokers, those who had quit for more than 5 years were also considered as having an ideal smoking status³³.”

Lines 332-338:

“Our study used a modified LS7 measure, with specific modifications made to the subjective LS7 factors, including diet, physical activity, and smoking status. These modifications aimed to more precisely capture the characteristics and lifestyle impacts on cardiovascular health in the BRHS. Previous BRHS reports have demonstrated associations between the modified LS7 factors and markers or risk of cardiovascular diseases^{28,29,33}. The diet measure EDI has been shown to be associated with the risk of CHD in the BRHS²⁸, and physical activity and smoking status have been found to be related to the markers of cardiovascular diseases in the BRHS^{29,33}.”

As mentioned in the previous response, we acknowledge the introduction of LE8; however, lacking sleep information in the BRHS study limited our ability to include sleep health measure in the current study. Given the high correlation of LS7 and LE8 (correlation coefficient $\rho = 0.88$, $P < 0.0001$) and the indistinguishable difference between the two metrics in their associations with the risk of CVD^{48,49}, excluding sleep metric may not affect the findings largely. We have included a discussion paragraph regarding the comparability of LS7 and LE8, in lines 340-350.

Manuscript References:

27. Kourlaba, G., Polychronopoulos, E., Zampelas, A., Lionis, C. & Panagiotakos, D. B. Development of a Diet Index for Older Adults and Its Relation to Cardiovascular Disease Risk Factors: The Elderly Dietary Index. *Journal of the American Dietetic Association* **109**, 1022–1030 (2009).
28. Atkins J. L. *et al.* High diet quality is associated with a lower risk of cardiovascular disease and all-cause mortality in older men. *J Nutr* **144**, 673–680 (2014).
29. Wannamethee, S. G. *et al.* Physical Activity and Hemostatic and Inflammatory Variables in Elderly Men. *Circulation* **105**, 1785–1790 (2002).
33. Wannamethee, S. G. *et al.* Associations between cigarette smoking, pipe/cigar smoking, and smoking cessation, and haemostatic and inflammatory markers for cardiovascular disease. *European Heart Journal* **26**, 1765–1773 (2005).
48. Lloyd-Jones D. M. *et al.* Status of Cardiovascular Health in US Adults and Children Using the American Heart Association’s New “Life’s Essential 8” Metrics: Prevalence Estimates From the National Health and Nutrition Examination Survey (NHANES), 2013 Through 2018. *Circulation* **146**, 822–835 (2022).
49. Howard, G. *et al.* Comparative Discrimination of Life’s Simple 7 and Life’s Essential 8 to Stratify Cardiovascular Risk: Is the Added Complexity Worth It? *Circulation* **148**, 00-00. CIRCULATIONAHA.123.065472 (2023).

The analysis only focused on CMD-free participants. It is well known that a variety of chronic diseases occur simultaneously and mutually reinforcing in the elderly and can contribute to impairing life expectancy. However, this study did not take into account other chronic diseases, such as dementia, cancer, chronic kidney disease, and so on.

Response: We appreciate the reviewer’s suggestion of including other chronic diseases. Nevertheless, our current study has been deliberately designed to focus specifically on cardiometabolic diseases, which are well-established as being associated with and caused by poor lifestyle choices. Our study is hence uniquely positioned to illustrate the potential positive impact of a healthy lifestyle on cardiometabolic health in the elderly.

Only social class, alcohol intake, and energy intake were adjusted in the model, but some important confounders such as education level, income, and other chronic diseases, which are associated with life expectancy should be controlled for.

Response: We thank the reviewer’s suggestion on adjusting for additional confounders. Unfortunately, the BRHS did not record information on education level or income at Q20. However, we have adjusted for social class based on occupation in our model as an indicator of socioeconomic position, which has been suggested to be a more important indicator of mortality than education level in the UK population¹. Moreover, we conducted an additional sensitivity analysis to examine the impact of further adjusting for the National Index of Multiple Deprivation (IMD) quintiles in the model, which took account of other socioeconomic factors, including income, employment, education, housing, and living environment at neighborhood level⁴¹. We found that the baseline IMD quintiles and social class were highly correlated (Chi-square test $p < 0.0001$), and the model results essentially remained unchanged following the additional adjustment. We have now included the result of additional adjustment for the IMD in Supplementary Table 4.

Regarding prevalent chronic conditions, our study recorded participants’ prevalent cancer status at baseline. There were 140 cancer cases (5.3%) reported at baseline. While cancer could potentially act as a confounder, given prior evidence suggesting that certain types of

cancer may have been the result of a poor LS7 profile², cancer could also be mediating the association of LS7 with CMD-free life expectancy. Nevertheless, to assess the impact of additionally adjusting for baseline cancer status, we conducted a sensitivity analysis. We found that further controlling for baseline prevalent cancer status did not change the model results. We have now included the results in Supplementary Table 5.

We have added the explanation of additional analyses in methods section, lines 201-207: “Additionally, several sensitivity analyses were conducted to further examine the robustness of our findings. These included an additional adjustment for the National Index of Multiple Deprivation (IMD) quintile, which is a socioeconomic indicator derived from baseline neighborhood-level socioeconomic factors such as income, employment, education, housing, and living environment⁴¹. We also assessed the impact of additionally adjusting for baseline prevalent cancer status, which may be a result of a poor lifestyle and affect participants’ life expectancy without CMD.”

References:

1. Smith, G. D. *et al.* Education and occupational social class: which is the more important indicator of mortality risk? *J Epidemiol Community Health* **52**, 153–160 (1998).
2. Van Sloten, T. *et al.* Association of Midlife Cardiovascular Health and Subsequent Change in Cardiovascular Health With Incident Cancer. *JACC: CardioOncology* **5**, 39–52 (2023).

Manuscript References:

41. Ramsay S. E. *et al.* The influence of neighbourhood-level socioeconomic deprivation on cardiovascular disease mortality in older age: longitudinal multilevel analyses from a cohort of older British men. *J Epidemiol Community Health* **69**,1224–1231 (2015).

Supplementary Table 4.

The Hazard Ratios (HRs) of the associations between Life’s Simple 7 adherence and risk of cardiometabolic diseases/death in BRHS participants aged 60-79 in 1998-2000, further adjusted for IMD quintiles (n=2662).

LS7 Adherence	HR	95% CI	P-value
Poor (Achieved 0-1 LS7 ideal levels)	Ref	-	< 0.01
Intermediate (Achieved 2-3 LS7 ideal levels)	0.74	(0.65, 0.84)	
Ideal (Achieved 4+ LS7 ideal levels)	0.59	(0.51, 0.70)	

IMD: National index of Multiple Deprivation (IMD).

Models used age as time scale, and adjusted for social class, alcohol intake, energy intake, and IMD. LS7 = Life’s Simple 7.

Supplementary Table 5.

The Hazard Ratios (HRs) of the associations between Life’s Simple 7 adherence and risk of cardiometabolic diseases/death in BRHS participants aged 60-79 in 1998-2000, further adjusted for baseline prevalent cancer status (n=2662).

LS7 Adherence	HR	95% CI	P-value
Poor (Achieved 0-1 LS7 ideal levels)	Ref	-	< 0.01
Intermediate (Achieved 2-3 LS7 ideal levels)	0.73	(0.64, 0.83)	
Ideal (Achieved 4+ LS7 ideal levels)	0.58	(0.50, 0.68)	

Models used age as time scale, and adjusted for social class, alcohol intake, energy intake, and baseline prevalent cancer status. LS7 = Life's Simple 7.

Reviewer #3 (Remarks to the Author):

The manuscript titled “Association of Life’s Simple 7 lifestyle metric with cardiometabolic disease-free life expectancy in older British men” investigates whether a composite lifestyle measure in older men is associated with risk and years of life free from cardiometabolic disease and death. This composite measure is one of many, which captures the participants level of adherence to 7 select lifestyle factors, ranging from smoking status, to physical activity and BMI. Here, the authors found that older men adhering to an ‘ideal’ lifestyle gained more years of CMD-free life than men who adhered to a ‘poor’ lifestyle. The authors also focused on each factor of the composite measure individually, and found physical activity to provide the greatest benefit in CMD-free survival compared to those men that were more inactive.

This was a well written study that confirms our general understanding of these lifestyle factors. For example, being physically active is better for your health than being inactive. While this is of value to the knowledge base, the authors description and interpretation of the composite measure weakens the importance of these findings. Interpreting such composite measures is a challenge, but is vitally important to what the reader can extract from the current evidence. My recommendation is to have the authors revise this aspect of the paper before it can be considered for publication. My specific comments are as following:

Response: We appreciate the reviewer’s comment on the challenge of interpreting composite measure. We have revised the manuscript accordingly based on the specific comments provided below. Please find our detailed response to each comment.

ABSTRACT

1. What is the Life’s Simple 7? Can the authors provide a brief comment regarding the factors included in the LS7, as this will help make sense of measures categories i.e., poor, intermediate or ideal, and the significance of these results

Response: We thank the reviewer for the suggestion. We have added details of the LS7 lifestyle factors in the abstract.

“Each LS7 factor (BMI, blood pressure, blood glucose, total cholesterol, smoking, physical activity, and diet) was categorized as poor, intermediate, or ideal, and a composite LS7 adherence was determined by summing the number of LS7 ideal levels achieved.”

2. Line 38: Can the authors include the percentage of participants with fatal or non-fatal CMD events e.g., 819 (31%)

Response: We have now included the percentage of participants who experienced CMD events and death events.

“During a median follow-up of 19.3 years, 819 (30.8%) fatal or non-fatal CMD events occurred, as well as 945 (35.5%) individuals who died from other causes.”

3. Line 41: “life gain of 4.37..... compared to men with the least Ls7 adherence” Can the authors include the number of years for those with the least LS7 adherence.

Response: We have added information on the number of CMD-free life years for participants with the least LS7 adherence [18.42 years (95% CI: 16.93, 19.90) at age 60] in the abstract results section:

“Compared to men with the lowest LS7 adherence [with 18.42 years (95% CI: 16.93, 19.90) of CMD-free life at age 60], men with an ideal LS7 adherence had an additional 4.37 years (95% CI: 2.95, 5.79) of CMD-free life.”

INTRODUCTION

1. Line 79: “Lifestyle factors are crucial modifiable risk factors for CMD...” can the authors provide a brief example e.g., physical activity reduces risk, and how this can be used to derive measures of overall lifestyle e.g., these can be used to derive measures of overall lifestyle.

Response: We thank the reviewer’s suggestion on providing brief examples of individual lifestyle factors before proceeding to overall lifestyle measure. We have added in the examples and revised the transitions, lines 78-84:

“Lifestyle factors are crucial modifiable risk factors for CMD. Specific lifestyle risk factors, such as obesity, physical inactivity, and smoking, have been largely studied and are associated with the development of CMD⁸. While each risk factor is important individually, it was suggested that lifestyle factors are inter-related and a combined healthy lifestyle may have synergistic effects in preventing diseases^{9,10}. There has been more research focusing on understanding the health impacts of a composite lifestyle, which is derived by summing up individual lifestyle metrics such as body mass index (BMI), physical activity level, and smoking status.”

2. Line 78: “Among lifestyle measures, the Cardiovascular Health (CVH)..... has been widely studied” I recommend the authors stick with CVH or LS7, not both.

Response: We appreciate the reviewer’s suggestion to stick with either CVH or LS7. We have now retained only the term “LS7” in the manuscript.

3. Line 85: “that an ideal CVH profile” can the authors give an example of what this entails e.g., “an idea CVH profile (i.e., lower BMI, blood pressure and greater physical activity) was significantly related to a lower risk of incident MI and stroke...”

Response: We have included the examples of an ideal LS7 profile in the sentence:

“Prior studies have consistently found the inverse association of LS7 with risk of CMD. A meta-analysis including 11 cohort studies showed that an ideal LS7 profile (i.e., lower BMI, BP, blood glucose, and total cholesterol, regular physical activity, healthy diet, and no smoking) was significantly related to lower risks of incident MI and stroke [HR (95% CI): 0.24 (0.15, 0.34) and 0.33 (0.21, 0.45) for MI and stroke, respectively] when compared to poor LS7 profile.”

METHODS:

1. Line 132: “To reflect adherence to the LS7 metric criteria, each LS7 metric was categorized as poor, intermediate, or ideal”. Are the authors able to link this sentence to the 7 factors categorized by the level of adherence. For example, “The level of adherence to each of these 7 lifestyle metrics was defined as poor, intermediate or ideal...”

Response: Following the reviewer’s recommendation, we have revised the relevant sentence to explicitly link the level of adherence to each LS7 lifestyle metric:

“The level of adherence to each LS7 lifestyle metric was categorized as poor, intermediate, or ideal.”

2. Lines 155-156: can the authors provide an example for each social class? For example, “non-manual (e.g., office work)....”

Response: We have provided several examples for each social class:

“Social class was classified into manual (e.g., bricklayers, bus conductors, general laborers), non-manual (e.g., engineers, teachers, clerks), or unspecified based on the longest held position coded by the Registrar General’s occupational classification.”

RESULTS:

1. Line 197-198: are the authors able to provide a breakdown of the CMD and death events, as per the abstract i.e., “819 fatal or non-fatal CMD events occurred, as well as 39 945 individuals who died from other causes.”

Response: We have added the breakdown of the CMD and death events in the results section:

“Of 2662 men being included in the analyses, the mean age at enrollment was 68.2 years. During a median follow-up of 19.3 years (Q1, Q3: 18.8, 19.9), 819 (30.8%) participants developed fatal or non-fatal CMD events, and there were 945 (35.5%) individuals died from other causes.”

2. Line 199: given a LS7 category of poor adherence is based on measures of high blood pressure, low physical activity, smoking and a higher BMI, we would expect these findings, and thus I see no real purpose in reporting these metrics. However, what the authors have written regarding work, alcohol, blood glucose and total cholesterol are interesting and should remain.

Response: We appreciate and agree with the reviewer’s suggestion of excluding expected LS7 characteristics. We now have revised the results paragraph:

“Men adhered the least to the LS7 were more likely to be manual workers and heavy alcohol drinkers, and had poorer LS7 metrics, including higher blood glucose and total cholesterol.”

DISCUSSION:

1. At the start of the discussion can the authors begin by reinstating what their primary aims of the study were, followed by an interpretation of their findings. Currently it reads like the results.

Response: We thank the reviewer’s suggestion on revising the first discussion paragraph. We have revised the discussion paragraph by reinstating the study primary aims and followed by interpretation of the findings:

“The current study aimed to quantify the number of years of life gained without CMD attributable to adherence to the LS7 lifestyle metric in the elderly and investigated whether social class modified the association. Our results revealed that, for older British men living an ideal composite lifestyle as recommended by the AHA LS7 lifestyle metrics (i.e., maintaining healthy BMI, BP, blood glucose, and total cholesterol levels, and regularly participating in physical activity, eating a healthy diet, and not smoking), at the age of 60, they could potentially gain more than 4 years of life without CMD, compared to those reporting a poor LS7 adherence. And this health benefit was consistent across social class groups. Notably, among LS7 lifestyle metrics, we found that having a moderate to vigorous physical activity level improved the CMD-free life expectancy the most, with an estimated of 4.84 years of life gain compared to participants who were physically inactive.”

2. Line 252: “Findings of the current study are consistent with previous studies” can the authors include references here.

Response: We apologize for not including references here. We have added the references followed the sentence in line 274.

3. Line 262: “...UK cohort study. Followed 341,331...” can the authors revise the start of this sentence e.g., “Following 341,331 participants for a median of 11.4 years...”

Response: We thank the reviewer’s suggestion. We have revised the start of the sentence to “Following 341,331 participants for a median of 11.3 years”.

4. Line 252-272: can the authors elaborate on the type of factors included in these composite lifestyle measures. While the last reference is most similar to this study, it would be interesting to get some general understanding of what type of composites equated some of these other results.

Response: We appreciate the reviewer’s valuable feedback. We have added specific details of the lifestyle factors included in the previous studies:

“Specifically, a prior study that involved 12 European cohorts computed a composite lifestyle score based on BMI, smoking, physical activity, and alcohol consumption¹⁷. The study found that individuals with the best composite lifestyle score had an average of 9.9 additional years of life without chronic diseases for men and 9.4 additional years for women between ages 40 and 75 years, compared to those with the worst lifestyle score. Another prior study including two large cohorts in the UK and USA showed that people could gain up to 12 years of life without chronic diseases at the age of 50 if having no behavioral risk factors (i.e., having a less than 30 kg/m² BMI, not smoking, being physically active, and consuming alcohol less than 5 days a week)²⁰.”

5. Line 274-284: can the authors provide a brief explanation as to why they think manual workers may have fewer years of CMD-free life compared to non-manual workers? What does the evidence suggest?

Response: We thank the reviewer’s suggestion. We have provided a brief explanation on the health expectancy difference between manual and non-manual workers, in lines 303-306:

“Potential explanations of the health disparity between manual and non-manual workers are that manual workers may have less favorable working conditions, limited healthcare access, and higher stress levels⁴⁴. This social class difference in healthy life expectancy warrants further clinical and public health consideration.”

6. Line 287: “ideal physical activity level” can the authors note what this means e.g., moderate activity per week etc...

Response: We have included the meaning of ideal physical activity level in line 315:

“We found that maintaining a usual pattern of moderate to vigorous physical activity, which indicated an ideal physical activity level, could potentially provide the largest CMD-free survival benefit of all components of the LS7 score.”

REVIEWERS' COMMENTS:

Reviewer #1 (Remarks to the Author):

Happy with the changes.

Reviewer #2 (Remarks to the Author):

The authors have largely answered my questions. I have no more suggestions.

Reviewer #3 (Remarks to the Author):

I would like to thank the authors for addressing all my comments. In my opinion, following a secondary review, the paper is now appropriate for publication.